# Temperature Sensor Based on Surface Plasmon Resonance with TiO_2_-Au-TiO_2_ Triple Structure

**DOI:** 10.3390/ma15217766

**Published:** 2022-11-03

**Authors:** Yutong Song, Meng Sun, Haoyu Wu, Wanli Zhao, Qi Wang

**Affiliations:** 1College of Sciences, Northeastern University, Shenyang 110819, China; 2Science and Technology on Electro-Optical Information Security Control Laboratory, Tianjin 300308, China

**Keywords:** surface plasmon resonance (SPR), TiO_2_-Au-TiO_2_ nanostructure, temperature sensor, Finite Element Method (FEM) simulation

## Abstract

Temperature sensors have been widely applied in daily life and production, but little attention has been paid to the research on temperature sensors based on surface plasmon resonance (SPR) sensors. Therefore, an SPR temperature sensor with a triple structure of titanium dioxide (TiO_2_) film, gold (Au) film, and TiO_2_ nanorods is proposed in this article. By optimizing the thickness and structure of TiO_2_ film and nanorods and Au film, it is found that the sensitivity of the SPR temperature sensor can achieve 6038.53 nm/RIU and the detection temperature sensitivity is −2.40 nm/°C. According to the results, the sensitivity of the optimized sensor is 77.81% higher than that of the sensor with pure Au film, which is attributed to the TiO_2_(film)-Au-TiO_2_(nanorods) structure. Moreover, there is a good linear correlation (greater than 0.99) between temperature and resonance wavelength in the range from 0 °C to 60 °C, which can ensure the detection resolution. The high sensitivity, FOM, and detection resolution indicate that the proposed SPR sensor has a promising application in temperature monitoring.

## 1. Introduction

Surface plasmon resonance (SPR) refers to the collective oscillation of electrons on the metal surface when incident light shines on the interface between metal and medium, and the coupling of light and electrons on the metal surface forms an electromagnetic wave propagating along the metal surface [1,2,3,4,5,6,7]. When the frequency of electron oscillation is consistent with the frequency of incident light, resonance is generated [4]. When resonance occurs, the electromagnetic field on the metal surface is enhanced. The SPR sensor made of surface plasmon resonance can subtly detect the change in the refractive index of the sensing medium. SPR sensors can be used to directly measure RI or indirectly detect any physical factor causing RI changes, such as temperature, concentration, strain, magnetic field, pressure, density, and molecular species [5,8,9,10,11,12,13,14,15,16,17,18,19].

A number of applications of temperature sensors can be seen in daily life, industrial production, and scientific research [20,21]. Sensors can monitor the room temperature and can also be used in air conditioners, induction cookers, microwave ovens, water dispensers, and other household appliances. In firefighting, measuring temperature is very important. Monitoring the temperature can find the abnormal temperature so that fire can be prevented, detected, and located [22]. In agricultural production, taking greenhouses as an example, suitable temperature is an important parameter for the plant growth processes [23]. In industrial production, such as metal smelting, the petrochemical industry, light industry, textile manufacturing, and water treatment, temperature sensors are the most common elements to ensure the normal operation of equipment [24]. In the medical industry, to ensure the activity of vaccines and other biological products, the environmental temperature should be constantly monitored during production and transportation [25]. From the above point of view, with the development of science and technology, research on temperature sensors has become a hot topic of research. The temperature sensor is developing in the direction of high accuracy and automation.

Due to the advantages of unmarkedness, real-time monitoring, quick response, and high sensitivity, SPR sensors also have important applications in temperature sensors in environmental regulation, food safety, medicine, and biological detection [8,26].

In metals where the SPR phenomenon occurs (such as gold, silver, aluminum, and copper), the precious metal gold is chemically stable and can output persistent SPR signals in the visible region [27,28,29,30,31,32]. Although gold has good oxidation resistance and corrosion resistance and can remain effective, the surface of the gold film is too smooth to adsorb a great number of molecules, thus limiting the improvement of sensor sensitivity. Titanium dioxide (TiO_2_), a nanostructured semiconductor metal oxide, has good absorbability, excellent chemical stability, loose molecular structure, wide band gap (i.e., 3.2 eV for anatase and 3.0 eV for rutile), and high RI (i.e., 2.5 for anatase and 2.7 for rutile) [33,34,35,36,37,38,39]. Owing to the above characteristics, the TiO_2_ layer is added to induce field confinement and enhancement in the interface, which is favorable for sensitivity improvement [40].

Changes in temperature can cause variations in the RI of alcohol, leading to the shift of the SPR spectrum. Nonetheless, the research on temperature sensors based on SPR film sensors has not attracted much attention. In this work, two SPR temperature sensors, i.e., TiO_2_-Au dual film structure and TiO_2_(film)-Au-TiO_2_(nanorods) triple structure, are proposed. The two sensors are simulated by the Finite Element Method (FEM). It is the purpose of this work to optimize the thickness of the TiO_2_ film and the Au film, as well as the geometry of the TiO_2_ nanorods, to maximize the sensitivity and FOM of the SPR temperature sensor. The optimized TiO_2_(film)-Au-TiO_2_(nanorods) triple structure SPR temperature sensor has a sensitivity of 6038.53 nm/RIU and performs −2.40 nm/°C in temperature sensing. The sensitivity of the proposed sensor is improved by 77.81% compared with that of a traditional gold SPR sensor. This work is important not only for the enhancement of the SPR sensor but for the study of temperature measurement by the SPR sensor as well.

## 2. Theoretical Analysis

SPR is the collective oscillation phenomenon of metal particles caused by photons when light irradiates the surface of metal materials at a specific angle. The necessary condition for exciting surface plasmon (SP) is that the wave vector of the polarized incident light ($k_{c}$) should be equal to the wave vector of surface plasmon ($k_{sp}$) [41,42].
(1)$$k_{c}=k_{sp}$$

The wave vector of the polarized incident light ($k_{c}$) can be represented as [43]:(2)$$k_{c}=\frac{2\pi }{\lambda }n_{p}sin\alpha$$
where $n_{p}$ is the RI of the prism, λ is the incident wavelength, and α is the incident angle.

The wave vector of surface plasmon ($k_{sp}$) is expressed as follows [44]:(3)$$k_{sp}=\frac{2\pi }{\lambda }\sqrt{\frac{n_{m}{}^{2}n_{s}{}^{2}}{n_{m}{}^{2}+n_{s}{}^{2}}}$$
where $n_{m}$ represents the RI of the metal film and $n_{s}$ represents the RI of the sensing medium.

In this paper, the wavelength interrogation method was adopted; that is, the incident angle was fixed, and the incident wavelength was scanned in a certain range. At a particular wavelength, the reflected light intensity sharply decreases, displaying a sharp valley in the reflectance curve. The wavelength corresponding to the minimum reflectance point is the resonance wavelength. The resonance wavelength moves gradually as the RI of the sensing medium changes. The refractive index of metals varies with wavelength, and the corresponding functional relationship is as follows [45]:(4)$$n_{p}sin\alpha =\sqrt{\frac{n_{m}{\left(\lambda \right)}^{2}n_{s}{}^{2}}{n_{m}{\left(\lambda \right)}^{2}+n_{s}{}^{2}}}$$
where *n_p_* is the RI of the prism, α is the incident angle, $n_{m}\left(\lambda \right)$ represents the RI of the metal film (varied with wavelength), and $n_{s}$ represents the RI of the sensing medium.

The Kretschmann configuration (Figure 1a), a common SPR structure, consists of a prism, a thin metal film, and the sensing medium. When there is only gold film, P-polarized light passes through the prism and undergoes attenuated total internal reflection at the prism/gold film interface. The evanescent wave penetrates the thin metal layer and resonates with propagating metal-dielectric surface plasmons, causing the absorption of the reflected light beam [26]. The electric field intensity is strongest at the intersection of gold film and sensing medium and declines exponentially with the increase in medium depth. After adding titanium dioxide, the high RI of TiO_2_ leads to the enhancement of the interaction of the evanescent field and shifts the resonant wavelengths towards the near-infrared wavelengths [46,47].

Sensitivity is an important parameter for evaluating the performance of SPR sensors and is generally calculated by the following formula:(5)$$S=\frac{\Delta \lambda_{SPR}}{\Delta n}$$
where $\Delta \lambda_{SPR}$ is the swift of SPR wavelength and Δn is the change in refractive index. The response sensitivity of SPR sensor to temperature is defined as:(6)$$S=\frac{\Delta \lambda_{SPR}}{\Delta T}$$
where ΔT is the change in temperature.

When it comes to evaluating sensor performance, sensitivity alone is not enough. The Figure of Merit (FOM) is another important factor, and the calculation is as follows:(7)$$S=\frac{\Delta \lambda_{SPR}}{\Delta T}$$
where *S* is the sensitivity of SPR sensor and FWHM is full width at half maxima of SPR reflectance spectrum.

In this work, sensors for four kinds of structures were simulated. The prism material is BK7 glass, and the sensing medium is ethanol liquid, whose refractive index is affected by temperature. The four kinds of metal layers are pure gold film, pure titanium dioxide film, TiO_2_-Au film, and TiO_2_(film)-Au-TiO_2_(nanorods) triple film, respectively. The angle of incident light is 72 deg. Temperature changes linearly with RI of ethanol liquid [48]:(8)$$n=1.36048-3.98\times 10^{-4}\left(T-T_{0}\right)$$
where *T* is the test temperature and *T*_0_ is the reference temperature (20 °C). Since ethanol is liquid at −144–78 °C, the performance of SPR sensors is explored at 0–60 °C (10 °C intervals), using the wavelength modulation method.

## 3. Results and Discussion

Firstly, the pure gold film SPR sensor was simulated. The sensor with a BK7 prism-Au thin film structure is shown in Figure 1b. The reflection spectra of Au SPR sensors with different thicknesses are shown in Figure 2a–f. The observations demonstrate that the SPR wavelength decreases as temperature increases. The SPR wavelengths show different shifts for different gold coating thicknesses. When the temperature increases from 0 °C to 60 °C, the SPR wavelength shifts are 95.1 nm, 86.5 nm, 81.1 nm, 77.9 nm, 75 nm, and 70.9 nm for the gold film thicknesses of 35 nm, 40 nm, 45 nm, 50 nm, 55 nm, and 60 nm, respectively.

In addition to the shift in resonance wavelength, the minimum reflectance should also be noted. With the increase in the gold film thickness, the minimum reflectance decreases first and then increases. The minimum reflectance at 0 °C and 60 °C is plotted in Figure 3. When the thickness of the gold film is between 35 nm and 60 nm, the minimum reflectance is higher than 20%. In general, sensors with a minimum reflectance greater than 20% are not considered, because the sensing effect is not good enough. In the subsequent research on sensor performance improvement, sensors with a minimum reflectance of above 20% will no longer be considered. SPR sensors based on gold film thicknesses of 40 nm, 45 nm, 50 nm, and 55 nm are explored in the following study.

Titanium dioxide is an oxide semiconductor and has the advantages of stable chemical properties and a high refractive index. Moreover, the molecular structure of titanium dioxide nanomaterials is dispersed and there is a large volume gap. These characteristics of TiO_2_ film make it more sensitive to refractive index changes than gold film. The prism structure with pure titanium dioxide film is shown in Figure 1c and SPR spectra are shown in Figure 4. It can be seen that the FWHM of the SPR curve of a pure TiO_2_ film SPR sensor is generally wide, up to 254.16 nm and 311.73 nm. In addition, there is no clear and definite SPR resonance wavelength. Therefore, TiO_2_ film alone is not a good choice.

In order to obtain the excitation effect of gold film and the enhancement effect of titanium dioxide at the same time, a prism-TiO_2_-Au SPR sensor was designed. The structure is shown in Figure 1d. In order to obtain the best combination of TiO_2_ layer and gold nanolayer, the simulation results in the sensing performance of different combinations of TiO_2_ layer thickness in the range of 140–190 nm (increasing 10 nm each time) and gold nanolayer thickness in the range of 40–55 nm (increasing 5 nm each time) are shown in Figure 5. As can be seen from Figure 5, in longitudinal comparison with the same gold film thickness, the sensitivity of the sensor first increases and then decreases with the increase in TiO_2_ layer thickness. This proves that the addition of titanium dioxide can indeed make the SPR wavelength move a longer distance and improve the performance, but if the TiO_2_ layer is too thick, the sensor’s response to the RI change begins to decline. Table 1 shows the detailed data of the six most sensitive material structure combinations.

The maximum sensitivity occurs when the SPR sensor is combined with 160 nm TiO_2_ and 40 nm Au, but due to the wide FWHM, the FOM is not so good. Based on the sensitivity and FOM, the SPR sensor combined with 160 nm TiO_2_ and 45 nm Au was finally selected as the sensor with the best effect. The SPR reflectance curve is shown in Figure 6a. In the temperature range of 0–60 °C, the SPR wavelength initially decreases as temperature increases. The SPR wavelength at 0 °C is 814.80 nm and is 691.00 nm at 60 °C, resulting in a resonance wavelength shift of 123.80 nm, with a sensitivity of 5184.25 nm/RIU. The minimum reflectance of the resonance wavelength is 0.032. Figure 6b shows the relationship between the temperature and SPR resonance wavelength, and the fitting is performed. Temperature and wavelength are linearly dependent, and the correlation coefficient R^2^ is 0.9754. The response of the sensor to temperature is −1.98 nm/°C.

To further improve the sensitivity, FOM, and linear correlation of resonance wavelength and temperature, titanium dioxide nanorods were added to form the TiO_2_(film)-Au-TiO_2_(nanorods) triple structure SPR sensor (Figure 7). In this structure, three factors need to be considered, namely the height, radius, and the distance between the nanorods. The influence of the above three factors on the sensing effect is presented in Table 2, Table 3 and Table 4.

To further study the performance of the sensor, the structure of TiO_2_ nanorods was changed, including height, radius, and spacing. The original geometric size of TiO_2_ nanorods is 50 nm in height, 15 nm in radius, and 20 nm in spacing. The control variable method was used in subsequent studies. In order to explore the influence of the height of TiO_2_ nanorods on the sensing performance (Table 2), the height was changed while keeping other parameters unchanged. It can be seen that the sensitivity of the sensor is improved while the height increases from 30 to 60 nm. When the height of the TiO_2_ nanorods is 60 nm, the sensitivity is the highest. However, the FWHM and FOM are poor at this time. When measuring the performance of a sensor, the sensitivity and the FOM should be considered comprehensively. As can be seen from the detailed data in Table 2, the FOM of the sensor is the largest and has a high sensitivity when the height is 50 nm.

Furthermore, the effect of the radius of the TiO_2_ nanorods and the spacing between the nanorods on the sensitivity of the sensor were investigated. The sensitivity of the sensor increases first and then slightly decreases as the radius increases from 5 to 20 nm. The value of FOM also increases first and then decreases, as shown in Table 3. When the radius of TiO_2_ nanorods is 15 nm, the sensitivity and FOM are both maximized.

In this study, the spacing range between TiO_2_ nanorods was 10–30 nm. With the increase in the spacing between nanorods, the sensor sensitivity shows a downward trend and the value of FOM first rises and then falls. Based on the data in Table 4, it can be concluded that the sensor with a 20 nm distance has the largest FOM and higher sensitivity and is the best choice.

In terms of geometric size, the optimized TiO_2_ nanorods in the sensor have a height of 50 nm, a radius of 15 nm, and a spacing distance of 20 nm. The sensitivity of the SPR sensor is 6038.53 nm/RIU. The reflectance curve is shown in Figure 8a. The SPR dip shifts towards shorter wavelengths with the increase in temperature. As shown in Figure 8b, the relationship between SPR resonance wavelength and temperature shows a good linear relationship with a correlation coefficient of 0.9990.

To compare the sensing performance of the pure Au SPR sensor, the TiO_2_-Au SPR sensor, the TiO_2_(film)-Au-TiO_2_(nanorods) triple SPR sensor, the SPR wavelengths, and their corresponding temperatures are presented in Figure 9. The sensitivity of the TiO_2_(film)-Au-TiO_2_(nanorods) triple SPR sensor is 16.48% higher than that of the TiO_2_-Au SPR sensor and 77.81% higher than that of the pure Au SPR sensor. This can be seen by comparing the slopes of the three fitted lines. In addition to the significant increase in sensitivity, the correlation coefficient of the TiO_2_(film)-Au-TiO_2_(nanorods) triple SPR sensor fitting line is also significantly higher than the other two fitting lines. In summary, the proposed TiO_2_(film)-Au-TiO_2_(nanorods) triple SPR sensor has high sensitivity and excellent FOM, with a good linear correlation between resonance wavelength and temperature.

To confirm the temperature sensing performance, the optimized TiO_2_(film)-Au-TiO_2_(nanorods) triple SPR sensor was simulated to determine whether it can respond clearly to temperature changes. Within the range of 10–30 °C, an SPR reflectance curve was calculated and drawn for every 1 °C, as shown in Figure 10a. The resonance wavelength at each temperature is plotted in Figure 10b and then fitted. The fitting results show that the resonance wavelength is linearly correlated with the temperature and the correlation coefficient is 0.9957. TiO_2_(film)-Au-TiO_2_(nanorods) triple SPR sensor can clearly distinguish the temperature change of 1 °C in the 10–30 °C range and the correlation is good. Therefore, when the resonance wavelength is known, the temperature can be determined according to the fitted curve.

Figure 11 shows the electric field |E| of the optimized TiO_2_(film)-Au-TiO_2_(nanorods) triple structure SPR temperature sensor at different interfaces of resonance wavelength when the temperature is 60 °C. When the reflectance is at its minimum, the intensity of the electric field approaches its maximum. In Figure 11, we can see the maximum electric field is obtained at the interface between the TiO_2_ nanorods and the sensing medium. The intensity field decays exponentially in the sensing medium. The electric field distribution indicates that the reduced reflectance is caused by the SPR phenomenon.

In the preparation experiment, TiO_2_ films can be manufactured via a sol-gel process [36,37], Au film can be manufactured via magnetron sputtering, and TiO_2_ nanorods can be manufactured via a hydrothermal method [49]. The optimized TiO_2_(film)-Au-TiO_2_(nanorods) triple structure SPR temperature sensor prepared through the above methods can obtain the theoretically expected sensing effect and can be put into application.

## 4. Conclusions

Temperature changes often cause variations in the RI of alcohol, thus leading to the shift of the SPR spectrum. However, the research on temperature sensors based on SPR film sensors has not attracted much attention. In this work, an SPR sensor with a TiO_2_(film)-Au-TiO_2_(nanorods) triple structure was proposed through a comparative study with a traditional pure Au film SPR sensor and a TiO_2_-Au dual film sensor. It was found that the triple combination of TiO_2_ and Au could not only excite SPR with Au but also enhance the performance with TiO_2_. The sensitivity of the SPR temperature sensor can achieve 6038.53 nm/RIU with 160 nm TiO_2_ film, 45 nm Au film, and 50 nm high TiO_2_ nanorods (5 nm in radius, 20 nm in spacing), and the detection temperature sensitivity is −2.40 nm/°C. The sensitivity of the TiO_2_(film)-Au-TiO_2_(nanorods) triple SPR sensor is 16.48% higher than that of the TiO_2_-Au SPR sensor and 77.81% higher than that of the pure Au SPR sensor. Good sensing characteristics show the application potential of the device in the field of temperature sensing. Owing to the high-temperature sensitivity, quick and clear response, simple structure, convenient operation, and environmental protection, the proposed TiO_2_(film)-Au-TiO_2_(nanorods) triple structure SPR sensor has great advantages in application.

## Figures and Tables

**Figure 1 materials-15-07766-f001:**
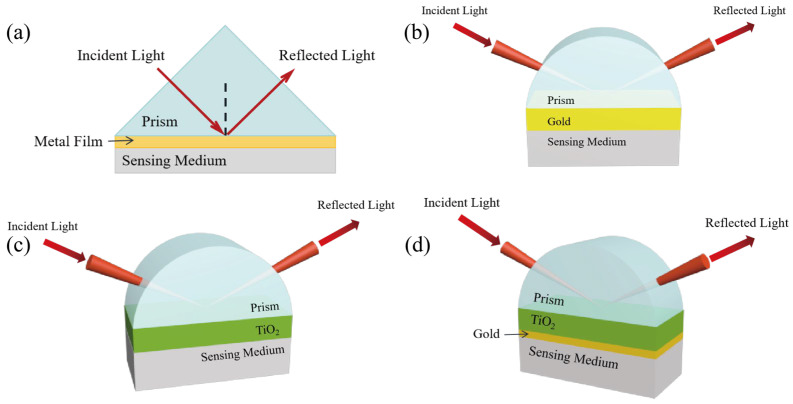
(**a**) Schematic diagram of Kretschmann configuration. (**b**) Schematic diagram of a pure gold (Au) SPR sensor. (**c**) Schematic diagram of pure titanium dioxide (TiO_2_) SPR sensor. (**d**) Schematic diagram of the proposed TiO_2_-Au dual-structure SPR sensor.

**Figure 2 materials-15-07766-f002:**
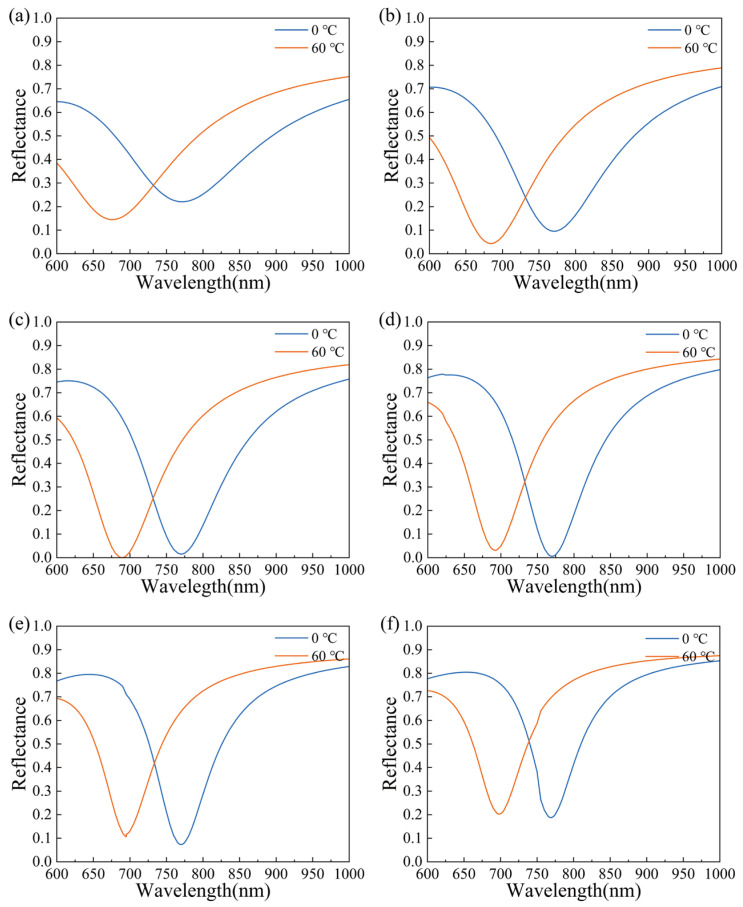
The reflectance curve of pure gold SPR sensors at different thicknesses of gold film: (**a**) 35 nm, (**b**) 40 nm, (**c**) 45 nm, (**d**) 50 nm, (**e**) 55 nm, and (**f**) 60 nm.

**Figure 3 materials-15-07766-f003:**
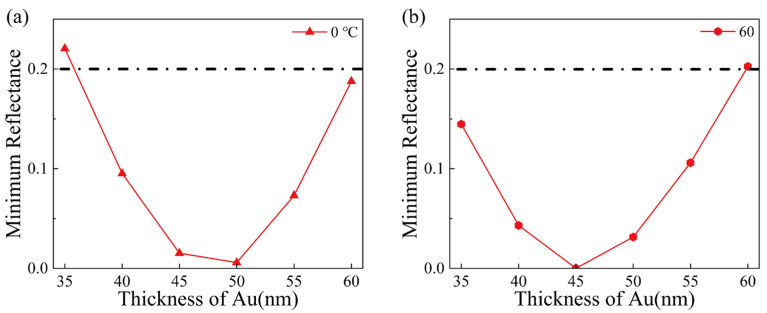
(**a**) Minimum reflectance of different gold thicknesses at 0 °C and (**b**) Minimum reflectance of different gold thicknesses at 60 °C.

**Figure 4 materials-15-07766-f004:**
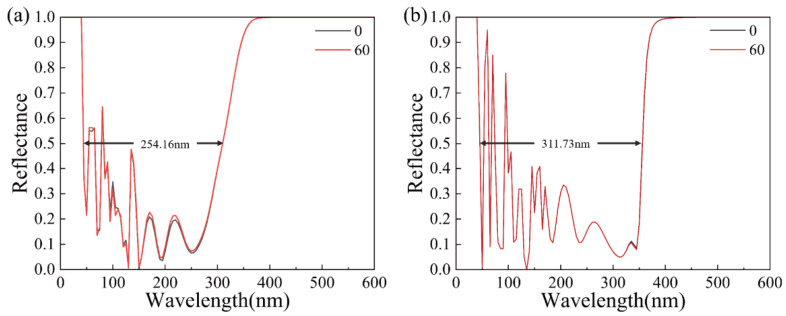
The reflectance curve of pure titanium dioxide (TiO_2_) SPR sensors at different thicknesses of TiO_2_ film: (**a**) 10 nm and (**b**) 160 nm.

**Figure 5 materials-15-07766-f005:**
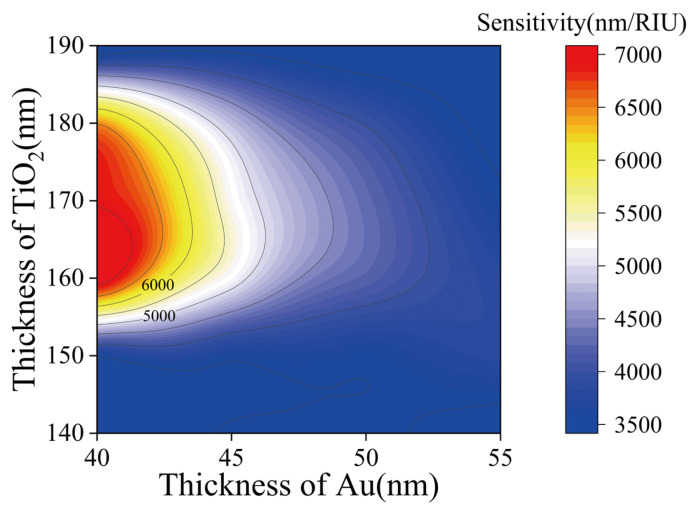
The RI response of the TiO_2_-Au SPR sensor at different thicknesses of gold film and TiO_2_ film.

**Figure 6 materials-15-07766-f006:**
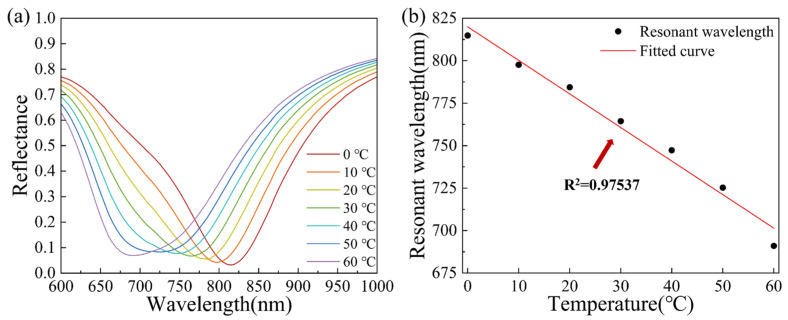
(**a**) The reflectance curve of the proposed 160 nm TiO_2_-45 nm Au SPR sensor in the range of 10–60 °C and (**b**) SPR wavelength corresponding to temperature of the proposed 160 nm TiO_2_-45 nm Au SPR sensor and linear fitting.

**Figure 7 materials-15-07766-f007:**
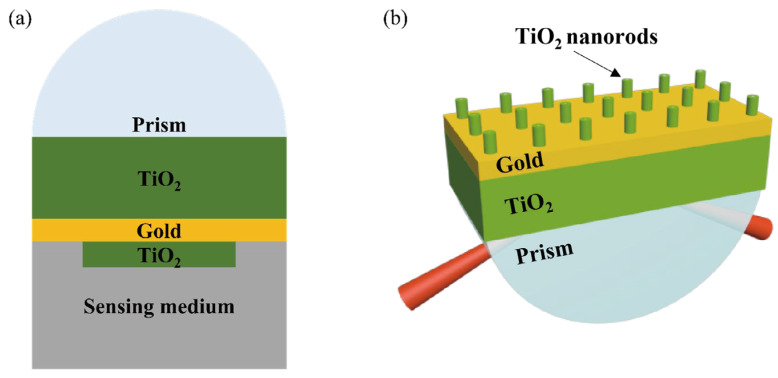
Schematic diagram of TiO_2_(film)-Au-TiO_2_(nanorods) triple structure SPR sensor (**a**) 2D (**b**) 3D.

**Figure 8 materials-15-07766-f008:**
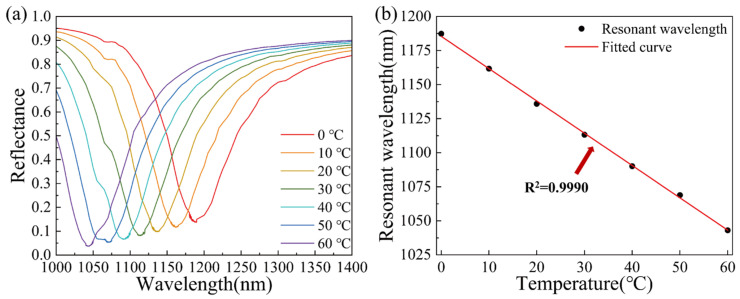
(**a**) The reflectance curve of the optimized TiO_2_(film)-Au-TiO_2_(nanorods) triple SPR sensor. (**b**) SPR wavelength corresponding to temperature of the optimized TiO_2_(film)-Au-TiO_2_(nanorods) triple SPR sensor and linear fitting.

**Figure 9 materials-15-07766-f009:**
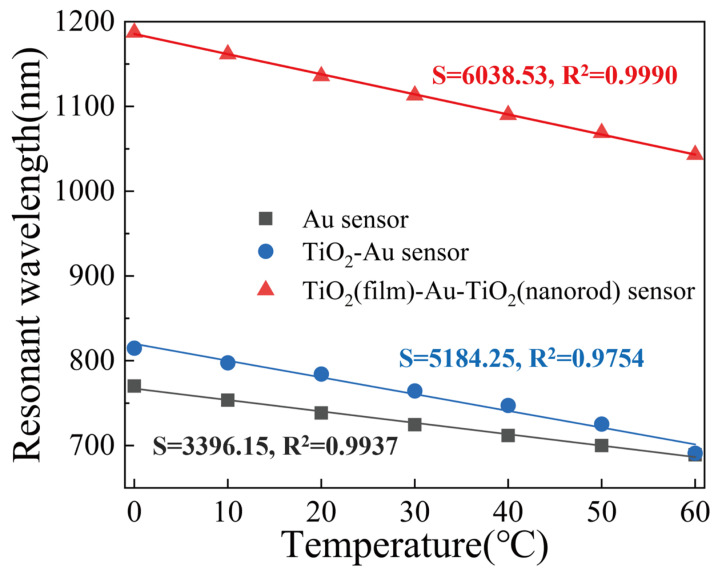
SPR wavelengths corresponding to temperature of three proposed SPR temperature sensors and linear fitting curves.

**Figure 10 materials-15-07766-f010:**
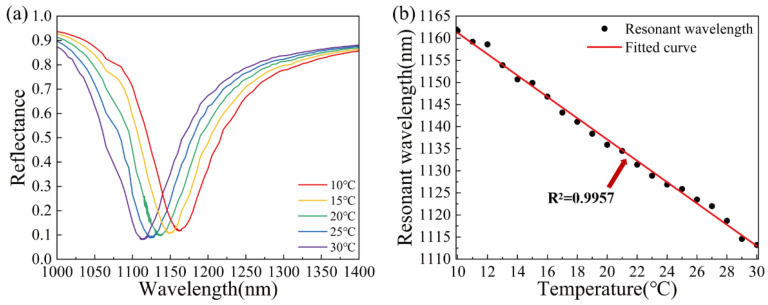
(**a**) The reflectance curve of optimized TiO_2_(film)-Au-TiO_2_(nanorods) triple SPR sensor in the 10–30 °C range and (**b**) SPR wavelength corresponding to temperature for every 1 °C in the 10–30 °C range and linear fitting.

**Figure 11 materials-15-07766-f011:**
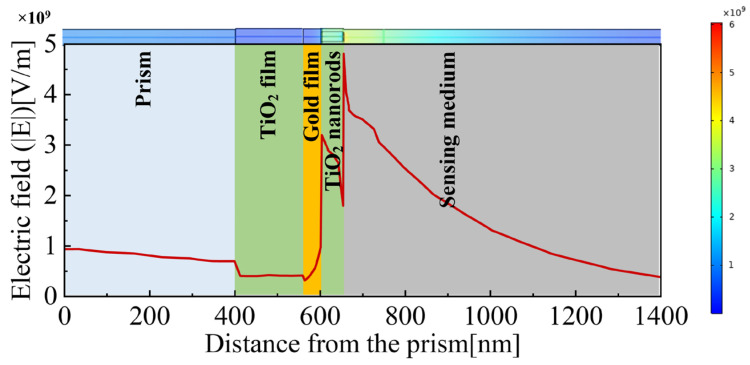
The electric field (|E|) distribution of the optimized TiO_2_(film)-Au-TiO_2_(nanorods) triple SPR sensor at resonance wavelength.

**Table 1 materials-15-07766-t001:** Sensitivity, FWHM, and FOM data of the TiO_2_-Au SPR sensor.

Thickness of TiO_2_ (nm)	Thickness of Au (nm)	ΔWavelength (nm)	Sensitivity (nm/RIU)	FWHM (nm)	FOM (/RIU)
160	40	172.4	7219.43	238.12	30.32
170	40	166.1	6955.61	214.92	32.36
180	40	155.0	6490.79	328.81	19.74
170	45	125.8	5268.01	158.39	33.26
160	45	123.8	5184.25	132.20	39.20
180	45	114.5	4794.81	176.06	27.23

**Table 2 materials-15-07766-t002:** Sensitivity, FWHM, and FOM data for different heights of TiO_2_ nanorods of TiO_2_(film)-Au-TiO_2_(nanorods) triple structure SPR sensor.

Height (nm)	ΔWavelength (nm)	Sensitivity (nm/RIU)	FWHM (nm)	FOM (/RIU)
30	111.0	4648.24	93.38	49.78
40	135.5	5674.20	92.91	61.07
50	144.2	6038.53	80.88	74.66
60	150.7	6310.72	103.8	60.8

**Table 3 materials-15-07766-t003:** Sensitivity, FWHM, and FOM data for different radiuses of TiO_2_ nanorods of TiO_2_(film)-Au-TiO_2_ (nanorods) triple structure SPR sensor.

Radius (nm)	ΔWavelength (nm)	Sensitivity (nm/RIU)	FWHM (nm)	FOM (/RIU)
5	102.3	4283.92	91.00	47.08
10	123.4	5167.50	93.21	55.44
15	144.2	6038.53	80.88	74.66
20	143.3	6000.84	108.03	55.55

**Table 4 materials-15-07766-t004:** Sensitivity, FWHM, and FOM data for different spacings between TiO_2_ nanorods of TiO_2_(film)-Au-TiO_2_(nanorods) triple structure SPR sensor.

Spacing (nm)	ΔWavelength (nm)	Sensitivity (nm/RIU)	FWHM (nm)	FOM (/RIU)
10	159.3	6670.85	98.04	68.04
20	144.2	6038.53	80.88	74.66
30	124.7	5221.94	96.51	54.11

## Data Availability

All the data are available in the main text.

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
