# Peer review of "Temperature Sensor Based on Surface Plasmon Resonance with TiO2-Au-TiO2 Triple Structure"

_materials, 2022, doi:10.3390/ma15217766_

Round 1
Reviewer 1 Report
Dear editor,
Temperature sensor based on surface plasmon resonance with 2 TiO2-Au-TiO2 triple structure, The paper, and the topic is quite interesting, but the results need to be improved. There are some missed understanding that requires some modifications in the paper. After the following corrections paper may be acceptable for publication:
1.The Abstract should be presented well especially highlighting the results.
2.The English and grammatical mistakes should be revised carefully.
3.This paragraph should be revised and correct some properties by adding the values for example the band gap reported…
[. Titanium dioxide 57 (TiO2), a nanostructured semiconductor metal oxide, has good absorbability, excellent 58 chemical stability, loose molecular structure, wide band gap, and high RI[35–37]. Owing 59 to the above characteristics, the TiO2 layer is added for inducing field confinement and 60 enhancement in the interface, which is favorable for sensitivity improvement[38,39]…
Also TiO2 have different applications
the suggested references;
https://doi.org/10.3390/nano12172901,
https://doi.org/10.1002/mame.200300067, https://doi.org/10.4995/Thesis/10251/160621, https://doi.org/10.1023/A:1011273700573
4.the authors could explain how The optimized TiO2(film)-Au-TiO2(nanorods) and if they manufacture TiO2 film and nanorods.
5. the FWHM of SPR curve of pure TiO2 film SPR sensor 150 is generally wide and there is no clear and definite SPR resonance wavelength. The authors could FWHM calculated and the information can be concluded from the values.
6. the authors should improve it and their lot of figures and without comment.
7. I recommend that the authors change Figure 6. (a) The reflectance curve of the proposed 160 nm TiO2-45 nm Au SPR sensor. It is not clear and 0 C is not appear
Reviewer 2 Report
Report on the manuscript materials-1997747 entitled “Temperature sensor based on surface plasmon resonance with TiO2-Au-TiO2 triple structure”.
The submitted manuscript should be revised. The following points should be addressed
1. The language of the manuscript should be revised.
2. To confirm the temperature sensing performance, the optimized TiO2(film)-Au- TiO2 (nanorods) triple SPR sensor is simulated to determine whether it can respond clearly to temperature changes within the range of 10-20℃. What about higher temperature 25 or 30 oC?
3. A section about the mechanism of sensor clearly indicating the role of Au and TiO2.
4. In the funding section, there is “Please, add” which should be removed.
5. figure 4 should be enhanced!
6. The conclusion part should be rewritten to be more scientific.
Round 2
Reviewer 1 Report
Dear Editor,
All the comments are addressed by the authors, and now the paper: 'Temperature sensor based on surface plasmon resonance with TiO2-Au-TiO2 triple structure' can be accepted for publication.
Keywords can be improved, the authors should cite equations 1 to 4 in the manuscript and Reference 40 is not correct, and there are mistakes in the author’s names; attached is the correct citation: Bouich, A. (2021). Study and characterization of hybrid perovskites and copper-indium-gallium selenide thin films for tandem solar cells (Doctoral dissertation, Universitat Politècnica de València) https://doi.org/10.4995/Thesis/10251/160621.
Reviewer 2 Report
The revised version could be accepted.
Author Response
I would like to take this great opportunity to thank you for the valuable comments. These comments have greatly improved the quality of this manuscript.